# A Novel Mobile Structured Light System in Food 3D Reconstruction and Volume Estimation

**DOI:** 10.3390/s19030564

**Published:** 2019-01-29

**Authors:** Sepehr Makhsous, Hashem M. Mohammad, Jeannette M. Schenk, Alexander V. Mamishev, Alan R. Kristal

**Affiliations:** 1Sensors, Energy, and Automation Laboratory (SEAL), Department of Electrical and Computer Engineering University of Washington, Seattle, WA 98109, USA; hashemm@uw.edu (H.M.M.); mamishev@uw.edu (A.V.M.); 2Fred Hutchinson Cancer Research Center, Seattle, WA 98109, USA; jschenk@fredhutch.org (J.M.S.); alan.kristal@fredhutch.org (A.R.K.)

**Keywords:** depth measurement, dietary measurement, 3d reconstruction, image segmentation, mobile structured light system, volume measurement

## Abstract

Over the past ten years, diabetes has rapidly become more prevalent in all age demographics and especially in children. Improved dietary assessment techniques are necessary for epidemiological studies that investigate the relationship between diet and disease. Current nutritional research is hindered by the low accuracy of traditional dietary intake estimation methods used for portion size assessment. This paper presents the development and validation of a novel instrumentation system for measuring accurate dietary intake for diabetic patients. This instrument uses a mobile Structured Light System (SLS), which measures the food volume and portion size of a patient’s diet in daily living conditions. The SLS allows for the accurate determination of the volume and portion size of a scanned food item. Once the volume of a food item is calculated, the nutritional content of the item can be estimated using existing nutritional databases. The system design includes a volume estimation algorithm and a hardware add-on that consists of a laser module and a diffraction lens. The experimental results demonstrate an improvement of around 40% in the accuracy of the volume or portion size measurement when compared to manual calculation. The limitations and shortcomings of the system are discussed in this manuscript.

## 1. Introduction

High accuracy estimations of the portion size component of energy intake are critical to nutritional research [1]. However, due to user bias, underreporting, and user burden [2], patients struggle to accurately estimate the volume of their food, even when assisted with established dietary recording methods [3,4]. Systems like the Remote Food Photography Method (RDPM) and a newly developed 3-dimensional (3D) scanning system attempt to minimize these limitations through the use of rapidly growing advancements in hardware and software [5].

The established and commonly used dietary measuring methods are complicated, time-consuming, and lead to around 60% accuracy [6]. This is especially visible in medical facilities when patients change their dietary habits to simplify the measurement process, which affects the overall process. A dietary measurement system completes the following three primary steps to calculate the nutritional intake of a meal: (1)Calculate the volume or estimate the portion size.(2)Identify the ingredients of the meal.(3)Convert the portion size information into nutritional data using information from steps 1 and 2.

Considerable work has been done to automate these steps using a visual sensor (camera). The general process uses a visual sensor to capture visual information from the food item. Once the visual data has been captured, the device needs to calculate the depth, which is the distance from the device to the food. The depth would allow the system to measure the volume or the portion size of the food. Table 1 shows a comparison between some of the common methods used for volume and portion size measurements. The system presented in this paper is called the Digital Dietary Recording System (DDRS), which tries to address some of the major shortcomings of traditional food volume and portion size measurement by finding the balance between increasing accuracy and reducing user burden. A comparison is made between the in-house testing and the reported results of each research team, when applicable.

This paper describes an alternative approach to dietary assessment using a structured light system to calculate the volume and portion size of a meal. The novelty of the presented system is to calculate the nutritional intake of a meal by constructing a 3-dimensional (3D) mapping of the meal in real time using a structured light system [11,12]. The algorithm can subsequently calculate the volume of the food using the 3D mapped model. 3D mapping is achieved by attaching a laser module that projects a matrix onto the surface of the food item. 

A 532 nm diode has been chosen instead of a 750 nm diode (infrared) because the 532 nm diode has the capability of reconstructing small 3D objects from a close range. The infrared spectrum is more common in larger 3D object reconstructions, which would not be as accurate when used on smaller objects such as a meal. Additionally, using the 532 nm (green) laser diode is easier for the user to align since the light is visible. The specific setup of the laser module add-on is shown in Figure 1. 

Using the smartphone application, the user will record a video of the meal. A scanning process that uses a 360-degree scan of the food will also be initiated. Once the smartphone application has recorded the video, the DDRS algorithm uses the structured light system, as well as the recorded video, to calculate the volume of the food [13,14]. The algorithm extracts the six most stable frames from the video captured by the user. The coordinate is then defined via a two-step process. First, the 3D coordinates are defined for each frame using the calibrated data. The calibration data provides the algorithm with two reference values: the depth or the distance to each laser dot and the degree of rotation based on the (0,0,0) reference point of the device position. Since the device is placed in a pre-set holder throughout the calibration process, it allows the algorithm to know the reference values for the X, Y, and Z axis of rotation. Second, once all six frames have been reconstructed in the 3D coordinates, the algorithm will estimate the rotational value, overlap the common spots from the edges for each frame to create a single coordinate, and reconstruct a full 3D model of the food item.

The process of using the DDRS device to record dietary intake (shown in Figure 2) is significantly simpler, faster, and more accurate than the existing dietary measurement tools. The DDRS device is used to take a 360-degree video of the food, which is then uploaded to the DDRS cloud database. This process allows nutritional analysts to instantly access and process the data using the DDRS algorithm and Nutritional Data Systems for Research (NDSR) [15]. 

The DDRS software is developed on the MATLAB (Matrix Laboratory, Natick, MA 01760-20980) computational platform and consists of three main algorithms: image segmentation, automatic laser dot detection, and 3D volume calculation. Figure 3 shows a flowchart for the DDRS software and how all the pieces connect to calculate the volume of a food item [16].

The structured light system enables the software to use each laser point as a reference to convert the pixel distance to the actual distance. This is critical for measuring the depth and overall dimensions. Before using the automatic laser dot detection algorithm to measure depth, the DDRS should be calibrated to create enough data sets to model the pixels to precise dimension measurements. Calibration is fully automated and is done when the device is assembled. 

The calibration process uses an automated system to take incremental measurements with a known depth value. Once the data is collected, the pixel distance to the actual dimension is generated by the conversion model. The conversion model is also used to calculate the distance of the phone from each dot and to create the depth measurement using this value. This measurement is based on the triangulation method, which is discussed in the upcoming section of this manuscript. 

This paper presents the structured light system calibration and automatic laser dot detection algorithm to measure the food volume and portion size. An explanation of the evaluation process is then provided with an analysis of the results from multiple experiments with different food samples. Finally, the effectiveness of the current implementation is discussed. 

## 2. Related Work

Considerable work has been accomplished in the field of dietary intake measurement and calculation. Some of the methods are well established in the field of nutritional research, such as 24-Hour Recall (24HR) [17]. 24HR is a questionnaire that enables users to capture detailed information about their dietary intake. 24HR is mostly used by researchers to understand eating habits. However, 24HR has shown inconsistencies and user burden due to inaccurate estimations and large time requirements. One of the major criticisms of existing food recording methods is that users often change the way they eat when they are recording. Thus, the records do not reflect the user’s “typical diet.” This is a common drawback to nutritional food diaries, which require users to manually measure (after a training process) their dietary intake and then write down this information into a food diary. 

Recently, more advanced and sophisticated methods propose using sensors, high-quality cameras, and image processing to reduce these limitations and burdens, as well as to increase the accuracy of the measurements. The recent improvements to mobile devices and their growing infrastructure has allowed researchers to use smartphones as their main recording tool, especially due to the smartphone’s accessibility and convenience of recording information. These methods generally estimate the volume of the meal using 3D modeling. 3D reconstruction is usually done in two common methods, single-view (using one image of the meal) and multi-view (using multiple images from around the meal). The multi-view feature, in which the user can take three images of the meal from different perspectives, has shown better results in capturing information of a meal from all angles. However, a method developed by a team at the University of Pittsburgh and Cheng Kung University uses a single-view method and estimates the volume based on the previously captured model-based register [18]. The advantages of this method include a decrease in scanning time and not requiring a reference object. The developed method contains three main sections: base plane localization, food segmentation, and volume estimation. The base plane localization is the stage in which the actual measurements are modeled based on the dimensions of the plate itself. This method requires the user to measure the diameter and depth of the plate prior to or after the scan. Overall, this method has shown a very low error rate; however, with irregularly shaped meals, the single-view method might be limiting the use of this system in real life scenarios.

DietCam, a recent method developed by a team at Michigan Technical University, uses a smartphone camera to calculate calorie intake [19]. The DietCam method contains an image manager, food classifier, and volume estimator that uses the 3D reconstruction of the food to calculate the volume. However, the DietCam method uses an external reference point and requires the user to calibrate the smartphone camera prior to the scan. The external reference object is a credit card, which allows the algorithm to calculate the actual dimensions of the meal. Use of a credit card limits the algorithm to capture an accurate calculation of the volume, which is satisfactory for general use, though unhelpful for users such as diabetic patients where results affect insulin prescription directly. 

Another advanced and commonly used approach, developed by a team at Purdue University, uses a colored index card to capture true dimensions [20]. This method is widely used in multiple nutritional studies. The index card is a 2-dimensional (2D) reference point, which disables the algorithm from capturing depth and texture for accurate volume measurements. In addition to the index card, to calculate the volume estimation of the food, this method uses a single image of the meal while factoring in the geometric properties using shape templates to reconstruct the 3D map of the food. The overall error rate reported for the known templates is reported at 11% for beverages and 8% for liquids. Despite the accuracy of the proposed method, it is limited to the database of shape templates.

Outside of dietary applications, SLS systems have been used in other 3D reconstruction and depth estimation applications. Recently, there has been some effort in using off the shelf SLS hardware, such as Microsoft Kinect, to reconstruct a 3D model [10,21]. Commercially available SLS hardware systems mainly use an infrared laser to get higher resolution by projecting more dots onto the object. Infrared lasers, which are used in the DDRS system, have many advantages over green lasers, such as higher resolution, higher accuracy, and better texture recognition. However, the use of an infrared laser is still limited to larger items, and it lacks accuracy in smaller items, such as mixed food plates. Recent developments by Occipital have resulted in a 3D scanner that uses an infrared laser which shows considerable promise with the DDRS algorithm in 3D modeling and volume estimation [22]. The current effort, which uses a structured light sensor developed by Occipital and the DDRS algorithm, is being prepared as a new article to cover the results of a pilot study conducted in collaboration with the Food and Nutrition Services at the Harborview Medical Center.

## 3. System Design

The DDRS is classified as a visual measurement system due to its visual sensor (smartphone camera) and a structured light system; the laser module is shown in Figure 4. DDRS uses a smartphone digital camera to project the 3D world onto 2D images [23]. To calculate the volume from a 2D image, the structured light system is used (shown in Figure 5) to measure two variables: the depth and the pattern (texture).

Figure 5 demonstrates the triangulation of a 3D point (shown in Figure 5). Once the system is calibrated (discussed in later sections), the relative distance between the projected point and the camera is determined using Equation (1):(1)$$\left|PO\right|=\frac{\left|OL\right|\times \sin(PLO{}^{\circ})}{\sin(LPO{}^{\circ})}$$

### 3.1. Design of the Structured Laser System for DDRS

The design of the SLS in the DDRS system follows the triangulation method shown in Figure 5 by using a laser module, angled mirror, and a diffraction lens; a customized enclosure is created using a Computer Aided Design (CAD) software called SolidWorks (shown in Figure 6) [24]. The enclosure is designed based on the calculations shown in Equation (1). To ensure that the laser projection is located in the center of the smartphone screen when projected, a 2-dimensional angled mirror is placed in front of the laser module as shown in Figure 6.

The laser module used to create the structured laser system is a 5 mW, 532 nm diode purchased from Laser Components [25]. Choosing a 532 nm laser diode allows better visibility for the user when operating the DDRS. Additionally, using a 5 mW instead of 1 mW module ensures enough power for the laser beam to pass through the diffraction lens and project onto the food items. A more detailed specification of the laser module is presented in Table 2.

The diffraction lens projects an 11 × 11 matrix onto an object, which allows for the calculation of depth and texture analysis once calibrated. Using a matrix grid laser allows the system to do a 360-degree scan of the food (shown in Figure 7) rather than only from the top angle of the plate. This allows higher accuracy by scanning the sides of the food item, which is not possible with other approaches such as a line laser. That being said, the matrix grid laser is limited in scanning the entire plate, which does not outweigh the higher accuracy due to the medical application of the system. There have been different approaches using the line laser instead, which enables auto calibration while reconstructing a 3D image while scanning the entire food [26]. Unfortunately, the line laser method is not suitable for this system due to its lower accuracy and limitation in 360-degree scanning.

### 3.2. Calibration Process Using Structured Laser System

The calibration process plays an essential role in the accuracy of the 3D reconstruction. Unlike other calibration methods that use the intrinsic and extrinsic parameters of the camera, DDRS calibration uses an Automatic Structured Light System Calibrator (ASLSC, shown in Figure 8). ASLSC includes a microcontroller, a servo motor, and a threaded pole to move the DDRS in precise intervals and take snapshots with high stability. The calibration process determines the depth map of a white plane at a known distance from the reference. This reference depth map aids in the construction of the depth map of other objects scanned.

To calibrate the device, we need to know how the laser dot pattern changes on a flat surface as we move the phone away from that surface. The simplest way to do this is by taking pictures of the laser grid at different distances from the phone. A white planar surface is used to project the laser grid and determine the pixel values of each dot in the image. Through calibration, a series of parallel and planar images within a valid range of depths are acquired from the camera, each containing a grid of dots. The pixel positions of the laser dots in the image change as a function of distance from the image plane, forming a line segment that can be mapped in 3D space. The steps required to generate the calibration data are as follows:Extract frames from the video: This process consists of converting video to frames, and determines which frames contain the entire laser pattern before the camera was covered. Next, only one image per delimiter is selected, and the results are saved with a name corresponding to the distance values.Locate the center of the laser grid and index the values of neighboring points: This process is done using the same automatic dot selection algorithm explained in this manuscript.Generate calibration data: This process calculates the depth or the distance from the phone to the center dot. In other words, the Z value is calculated here.

After extracting all the dots from the images, the pixel coordinates are processed to generate a map that will represent each dot as a field of a two-dimensional array. The idea is to find the distance from each point to the center. The smallest distance will be at the center point, so its assigned index is (0,0). The next four closest points are used to define the four quadrants (−1,0), (1,0), (0,−1), (0,1). Using these quadrants, we can map each point to an index, which will represent how far each point is located from the center.

The last step of 3D reconstruction is to calculate the real-world x and y values of the point cloud. The x and y values from which the reconstruction starts are the pixel locations of 2D points in the image, which must be converted to real dimensions in centimeters. The procedures to calculate either x or y values are very similar. The procedure for calculating x will be discussed first. Since the distance from the camera to the center dot is known, the farthest x (horizontal) distance can be calculated using Equations (2)–(5):(2)$$x_{ratio}=(x_{C}-\;\frac{x_{d}}{2})/(\frac{x_{d}}{2})$$
(3)$$x_{E}=tan\left(\frac{\theta }{2}\right)\cdot \left(z_{E}\right)\cdot \left(x_{r}\right)+C_{x\;shift}$$
(4)$$y_{ratio}=(y_{C}-\;\frac{y_{d}}{2})/(\frac{y_{d}}{2})$$
(5)$$y_{E}=\frac{2}{3}tan\left(\frac{\theta }{2}\right)\cdot \left(z_{E}\right)\cdot \left(y_{r}\right)+C_{y\;shift}$$
where $x,y_{C}$ is the x,y pixel values of the current laser dot, $x,y_{d}$ is the x,y (horizontal and vertical) dimensions of the picture, $z_{E}$ is the measured depth using the calibration process, and $x,y_{E}$ represent the estimated x,y coordinates.

The results of the calibration process show an accuracy of over 98%, and since the entire process is automated, the consistency is at 100%. Figure 9 shows the result of mapping out an entire 3D map of a single calibrated system.

### 3.3. Laser Dot Detection

The presented approach, which uses a laser dot detection algorithm on a segmented image, adapts the computation of several successive image filtering and masking operations. The process allows the algorithm to see the laser dots and capture the actual distance from each point [27]. The three primary filters implemented for the algorithm are: the HSV-controlled filter that intensifies the green dots, the cosine similarity masking of the two image pairs, and the luminance erosion of the object image. Figure 10 shows the high-level block diagram of the automatic dot detection, which consists of the following four main functions: HSV filter, cosine similarity, luminance erosion, and mask merge.

#### 3.3.1. HSV Filter

An HSV filter is first applied to the image to intensify the brightness of the green pixels. This filter comprises two separate filters: one filter mask that uses the hue component of the image to identify the colors of the pixels and another filter mask that uses the saturation component to distinguish the color intensities of the pixels. 

A hue filter is applied to mark only the pixels that are identifiable as the green pixels from the image. This highlights the laser dots placed on the food while the remainder of the image does not change. The computations used for calculating the hue and saturation values utilized for this filter are shown in Equations (6) and (7):(6)$$GH=90^{\circ }\leq PH\leq 180^{\circ }$$
(7)LCI=PS≥90%
where GH is green hue, PH is pixel hue, LCI is laser color intensity, and PS is pixel saturation. Once the HSV filter is applied, the image is converted from the HSV domain back to the red, green, and blue (RGB) domain to be processed by further filtering and masking techniques.

#### 3.3.2. Cosine Similarity Mask

At this stage, cosine similarity masking is used to isolate the laser dots further in the image. The approach begins by removing most of the background of the image (non-laser dots). This process is mathematically computed by determining the similarity of every pixel in the image to the determined color of the laser dots (filtering mask image); this is denoted by Equation (8).
(8)$$S=\cos\left(\theta \right)=\frac{A\times B}{\left[\left|A\right|\times \left|B\right|\right]}$$
where S is similarity.

As a starting point, the similarity of each pixel with respect to the color of pure green (RGB value of 0/255/0) is determined using Equation (8). This can be visualized as a 3D projection of the two vectors in the RGB domain on a color plane.

The computation and thresholding of the similarities between the two images create a binary mask that only marks pixels that are similar in both pictures (the bright laser dots). Depending on the brightness of the image and the reflection of the dots on the object, an iterative process may be executed to determine the color of the dots dynamically. 

#### 3.3.3. Luminance Mask

A combination of the cosine similarity mask and a luminance mask is used to achieve more accurate results by taking the intensity of the light into account. The two masks are applied separately to the copy of the original image; then, the two copies are multiplied to produce a result with more accurate dot locations.

The luminance filter begins by first isolating the background of the image, which, in this case, includes everything that is not a laser dot. This is accomplished by performing a morphological opening operation using the structuring element of a disk shape. In this scenario, the morphological opening means the dilation of the erosion of a set, A, by a structuring element, B. This operation is shown in Equation (9):(9)A∘B=(A⊖B)·B
where A is the image, B is the structuring element of a disk, ⊖ represents erosion, and ⊕ represents dilation.

The next step is to reduce the overall noise in the image, which is done by converting the image to a luminance matrix and thresholding the acceptance value to retain the brightest parts of the image. The resulting image contains laser dots and some of the borders without removing the noise comprised of the darker pixels. The luminance calculation is shown in Equation (10):(10)L=0.2126R+0.7152G+0.0722B
where L is the luminance, R is the red color value, G is the green color value, and B is the blue color value. This equation represents the luminance using the BT.709 HDTV standard [28]. The mask of this stage removes noise from the image while maintaining the laser dots and some points on the border. 

#### 3.3.4. Eliminating the Border and Applying the Masks

Eliminating the edges of the food (shown in Figure 11) requires image erosion which is applied with a structuring element to remove the thin outside edge on the border. Since the size of the structuring element is static, some dots on the border might be eroded and not located. The result is shown in Figure 11; the poor visibility of the green dots, shown in image Figure 11b, is improved by passing the frame through the luminance mask to overcome this hurdle, shown in image Figure 11c. 

#### 3.3.5. Eliminating Reflections

With the combination of the masks, there is now an accurate representation of the laser dots in the image. Using a visible laser beam creates a reflection, causing the camera to see more than one laser dot in the 2D image. 

To avoid these reflections, the algorithm measures the pixel distance between the reflected dots and selects the center one as the actual laser dot. Another aspect of the dot detection algorithm is locating the center dot, which is critical for the 3D reconstruction as can be seen in Figure 12, where the center laser dot is brighter than the other dots; this allows for the calibration of all other dots with the center dot, using it as a reference point. This brighter dot also helps the users sufficiently position the projected laser dots onto the food while scanning.

#### 3.3.6. Final Error Catching

A manual dot selection algorithm is created to ensure that there are no unselected or wrongly selected dots, the manual dot selection algorithm is implemented using MATLAB. This feature includes a Graphical User Interface (GUI) that allows the user to manually identify the laser dots, which are not detected by the algorithm. This final error-catching step can be seen in Figure 12. The use of celery is intentional to show the system’s limitation in identifying the laser dots on a green food item. The manual dot selection addresses this limitation.

The algorithm needs to make the estimated values to address unknown spots created in the shadow of the camera due to triangulation. Since not all spots are visible on the food, the algorithm looks for the closest available spot. For example, if the algorithm is missing a spot from the surface of the food, it measures the pixel distance between the available spot in both vertical and horizontal axes to the first available spot. If there is a spot available on the bottom of the plate, then an average distance is calculated to compensate for the missing spot. If there are no spots available, then the user is required to redo the scanning process. From experiments, this is a big limitation for the system when it deals with high-volume low-density food items, such as potato chips. In future iterations of the system, there will be attempts to address this issue, such as using a regression analysis to have a better estimation for such cases. Using the calibration data and the gyroscope data captured by the gyroscope sensor on the smartphone, the algorithm can identify the rotation information for each captured frame. To ensure consistency, the instruction for the scanning process asks the user not to rotate the device. These shortcomings are addressed in a future manuscript, which is near completion.

### 3.4. Volume Measurement

Finally, the volume measurement algorithm consists of extracting the most stable frames from each scan and then passing them through the segmentation and laser dot detection algorithms to get the 3D cloud point positioning using the calibrated data. Once each frame has been processed, using the photogrammetry, accelerometer, and calibration data of the distances of laser matrix to the DDRS camera, the overall volume can be calculated by placing all frames in a single 3D cloud point. 

Our current approach to finding the most stable frames is using the accelerometer data from the smartphone and extract the appropriate number of frames to cover all corners and keep the processing time under five minutes. Once the frames are extracted, each frame will be processed separately and combined using photogrammetry to create a 3D cloud point of the entire portion size. Figure 13 shows an overview of the volume measurement algorithm of the DDRS.

### 3.5. Frame Extraction

To achieve a thorough 360-degree scan of a meal, the user takes a circular video of the food. Since processing an entire video file would take hours, specific frames are chosen automatically by the algorithm using the accelerometer data to find the most stable frames. As seen, the highlighted points in Figure 14 are the frames which the highest stability coefficient can be achieved in all three axis. Based on the preliminary data calculations, the equilibrium number of frames for producing an accurate measurement and keeping the process time under 5 min, is six frames. Figure 14 shows the accelerometer data from one of the DDRS devices and the most stable timestamps.

## 4. Experiments

A pilot study has been done for testing the system usability at FHCRC. We have developed three functional prototypes of the mobile client. The purpose of the pilot study is to test how the participants accept the mobile client design. Ten participants attended the validation study. The demographics of the participants are as follows: five men and five women, five under the age of 40 and five over the age of 40, and ninety percent highly educated (high school or equivalent n = 1, college degree n = 5, graduate or professional degrees n = 4). Most of the participants were involved with some level of food shopping, preparation and meal planning. Each participant got the device and recorded a 3-day food diary using one prototype. We have developed a detailed user menu on how to use the device. Before the participants started the study, the nutritionist also gave them training on how to record data. The results of this experiment are presented in the Section 5.

## 5. Results

This section presents and analyzes the current results of running the algorithm presented in the approach section. The initial tests were conducted to evaluate the accuracy of the algorithm in detecting laser dots as well as segmenting the target food item. Figure 15 shows the results of the dot locations and segmentation on a small subset of sample images used for initial testing. Irregularly shaped items were used in the tests to analyze the cropping feature better. 

A set of tests were run to compare the current algorithm to manually selected laser dots. This would confirm improvements in both speed and accuracy. The results in Figure 16 show the improvement in all cases, especially in red dominant food items.

Figure 16 shows the performance of the DDRS algorithm when comparing the calculated volume using the laser dot detection algorithm integration and measured volume to the highly accurate water displacement technique. All items presented in Figure 16 have been computed using only the laser dot detection algorithm.

The results of the automatic laser dot detection algorithm show an average of 70% accuracy in the detection of the laser dots. This number is achieved by comparing over 180 frames; each frame is processed through the algorithm and then reviewed by individuals to monitor how many correct detections are made. However, the results have shown correlation between the color of the object and the accuracy of the algorithm. From the results, objects with darker colors (brown, black, etc.) and green color are more difficult to detect with the algorithm. In addition to the laser dot detection results, the segmentation algorithm shows a similar pattern of accuracy with a higher error in white color objects; this issue is due to the similarity of the plate color and the object. While changing the plate to a different color other than white would help with segmenting white objects, it does not address the average since color similarity is true for different color food items. To address both the segmentation and laser dot detection with specific color items could potentially be solved by using an infrared laser. Another limitation of this device is that the algorithm is not capable of calculating the bottom surface of the food on the plate. To address this limitation, the algorithm estimates the bottom surface by using the lowest visible spot on the food or plate from the side scans to reconstruct the volume. In a case where the food items are touching each other, and the bottom of the plate is not visible, the algorithm uses edge detection to estimate the top surface and extrudes that to estimate the total volume of the food item. The edge detection uses RGB values to identify the edge between two food items.

Additionally, another source of error in volume estimation comes from the low-density high-volume foods, such as stacked chips and scattered chips were also relatively high. This is due to the irregular shape of the object that creates random air gaps in between, hence, a large volume. This is partially addressed by conducting a 360-degree scan. The DDRS scanner uses 360-degree scanning approach to get information from the side angles of the food. This enables capturing more information such as the number of items (potato chips, grapes, etc.). The next step taken to address this issue is to approximate the volume adjustments by calibrating the algorithm to known low-density high-volume items. Unfortunately, this approach is limited since it is not possible to calibrate for all possible scenarios; however, since the system is designed for medical applications, often, the food items in the studies are known or discussed before. Hence, the calibration can be done to overcome this obstacle.

The overall percentage of error for the items tested is around 11% (shown in Figure 17). Compared to the estimated accuracy of the current solutions in medical dietary measurements, the DDRS has improved the dietary measurement by almost 40%. However, these tests were conducted by in-house using volunteers. To test the actual performance of the algorithm, a larger study is needed to evaluate all possible scenarios. 

Additionally, the volume estimations were converted to calorie calculations using the NDSR database, which converts the volume to weight using density. The results were compared to the actual measured weight of each item, which showed a higher error rate of almost 15%. There are two possibilities for this increase in error; (1) the NDSR library uses the USDA database, which is accurate; however, the nutrient ratio of volume to mass has not been explored in depth, since the use of volume estimation devices in dietary measurement is a new topic. (2) The error is created by multiple conversions, measuring weight by a scale is much more accurate due to its minimum conversions.

Finally, the last result is to look at the accuracy of the 3D reconstruction of the items. However, since the algorithm is mainly in charge of the volume estimation, the tests done on the 3D reconstruction were limited. The 3D reconstruction evaluation consists of testing ten plastic objects including plastic fruits. The 3D reconstruction results were compared to a fixed 3D scanner using the line laser developed by Illionix LLC (Seattle, WA, USA), shown in Figure 18), which is currently being used by the National Institute of Health and the Fred Hutchinson Cancer Center in a collaborative study.

The results from the initial pilot study with the Fred Hutchinson Cancer Center showed an improvement of 40% when participants used the DDRS compared to manual entry. The main sources of the improvement are eliminating the human error. When asked to use the manual food journal method, users were quickly overwhelmed with all the required measuring requirements. This resulted in misreporting and inconsistent of skipping some of the measurement. Also, the DDRS system enabled a much faster processing time, which allowed users to stick to their regular dietary habits. Change of dietary habits due to the complexity of dietary measurement methods is one of the leading shortcomings of manual dietary techniques.

The overall average error of the DDRS compared to the line laser showed an improvement of over 10%. The error rate is calculated by looking at the volume and the 3D point cloud comparison. The results of the scans are shown in Figure 19 and Figure 20. The DDRS scanner allows for 360-degree scanning, which can show its benefit compared to the line scanner. The DDRS scanner enables scanning of the object from all angles. Having that said, the remaining challenge is the reconstruction of the bottom surface, which is not visible by the camera. Unfortunately, the DDRS can only estimate the surface as flat between the visible laser dots on the edges of the food item. This creates uncertainty when there is air trapped under the food, such as soft solid foods.

## 6. Discussion 

Previous studies that use traditional methods show inaccurate measurements of portion size, even when the receipts and ingredients are given. The novelty of the DDRS system is in its ability to automate the process of measuring portion size by designing a portable structured light system as a smartphone hardware add-on. The structured light system approach is independent of the cooking method used to prepare the meal. The DDRS smartphone application requires the user to input the ingredients for one serving size of the meal or its recipe. The application also uses a crowdsourced library similar to the other commercialized applications, such as MyFitnessPal, for rapid recipe lookup [29]. 

A more comprehensive study is needed to test the performance of the device better and understand its compatibility in various cases. At this stage, the DDRS is still in the development phase and requires further improvements to be used in real life scenarios. However, with future funding, the device can be used in a research environment. The preliminary results show a promising path for the ease and use of the structured light system in the dietary measurement devices.

Using a laser module in the DDRS device shows it can improve the accuracy of the volume estimation; however, it also introduces a safety hazard for the user. The use of lasers is not recommended because they are harmful to a naked eye; however, the power rating of the laser module used in the DDRS is within the industry safety standard for consumer products. Regardless, the user should take cautionary steps, such as avoiding looking at the laser module directly and avoiding pointing the laser at another person.

### Future Work

DDRS could be significantly simpler and more efficient at measuring dietary intake than traditional methods by automating the portion size measurement. Moreover, such a system could help users stay consistent with their dietary habits. However, to establish the actual performance of the algorithm, multiple studies with larger and more diverse sample sizes need to be conducted. The initial large test samples, which were undertaken by the Fred Hutchinson Cancer Research Center, showed us that the system has a steep learning curve due to technological limitations, such as sensors, camera quality, and image processing.

Further large-scale studies could help us in analyzing the results and comparing them to empirical results found by the other more traditional methods. The future work of the DDRS algorithm will include making progressive values for the thresholding in the masks and HSV, instead of using static values. Furthermore, additional masks will be explored. It would be beneficial to test more objects with different colors and shapes for a deeper analysis of the algorithm by applying other variables, such as ambient light and multiple background colors.

## 7. Conclusions

This paper presented an algorithm to locate the laser dots in a segmented image containing food. The algorithm consisted of filtering via two masks: an HSV filtered cosine similarity mask and a luminance mask. The combination of the two masks, as well as the threshold filtering, isolates the laser dots as presented in the results section. An additional manual error-catching GUI was developed to ensure that the correct values are being passed to the 3D reconstruction phase of the DDRS. Various suggestions were presented in the analysis section for possible improvements to the algorithm.

## Figures and Tables

**Figure 1 sensors-19-00564-f001:**
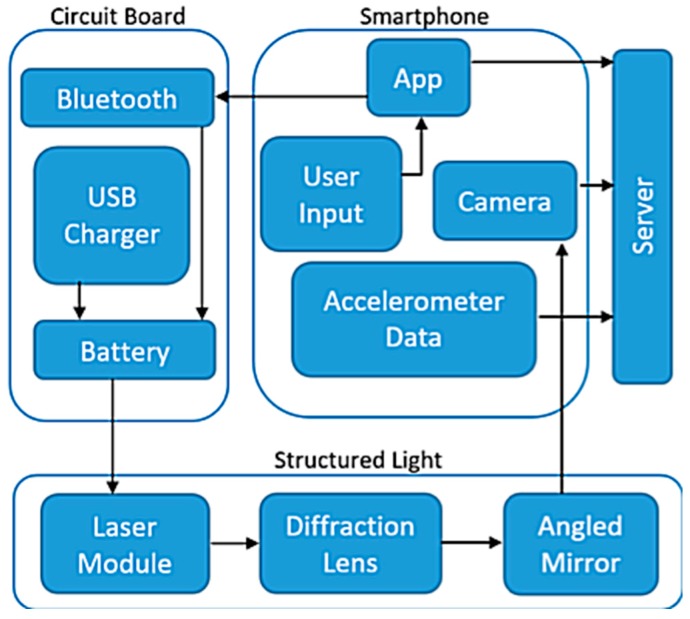
The structured light system designed for the DDRS device as a hardware add-on for the smartphone.

**Figure 2 sensors-19-00564-f002:**
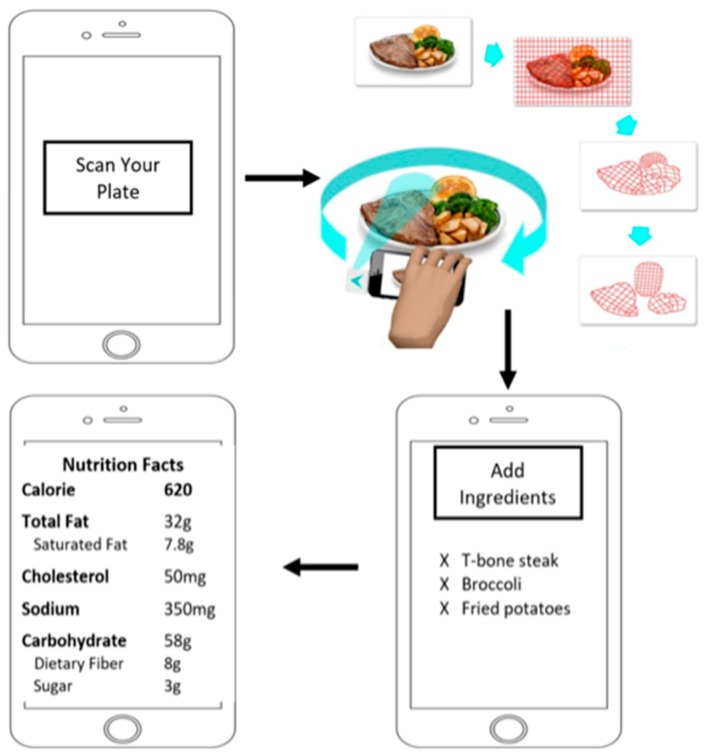
The overall DDRS system and the smartphone application. Starting from the top left, once the user starts the scanning process and the ingredients are added, the nutritional facts are generated based on the amount of food scanned.

**Figure 3 sensors-19-00564-f003:**
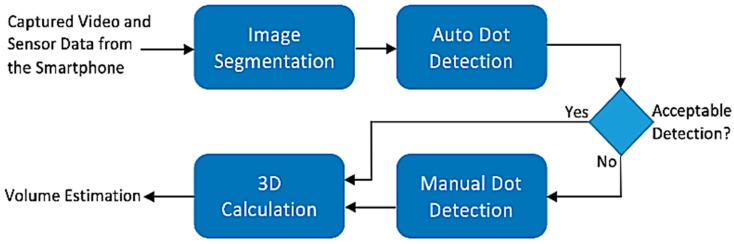
The DDRS volume calculation software, which shows the higher-level architecture of the algorithm. The algorithm uses the extracted frames and motion sensor data to isolate the image, calculate depth, and finally reconstruct a 3D model.

**Figure 4 sensors-19-00564-f004:**
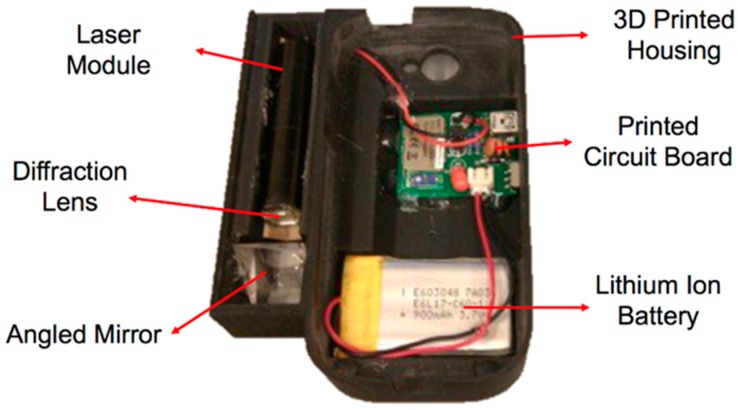
The DDRS enclosure, which holds the hardware add-on to power the laser module and communicates with the smartphone application via Bluetooth, the angled mirror to project the laser downward with the center dot visible at the center of the smartphone screen with the structured light system.

**Figure 5 sensors-19-00564-f005:**
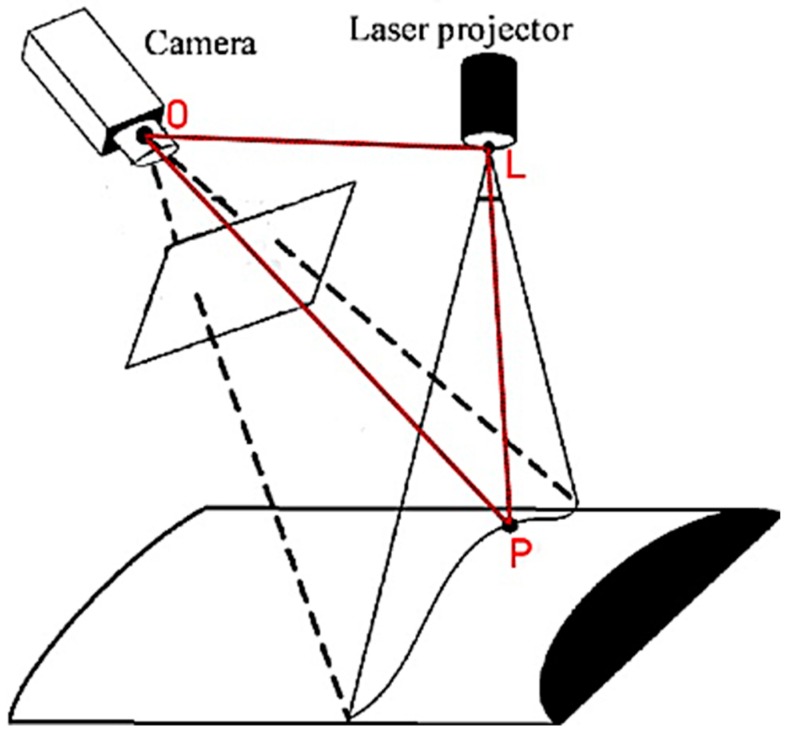
Triangulation method utilized in a structured light system for capturing the depth and texture [21].

**Figure 6 sensors-19-00564-f006:**
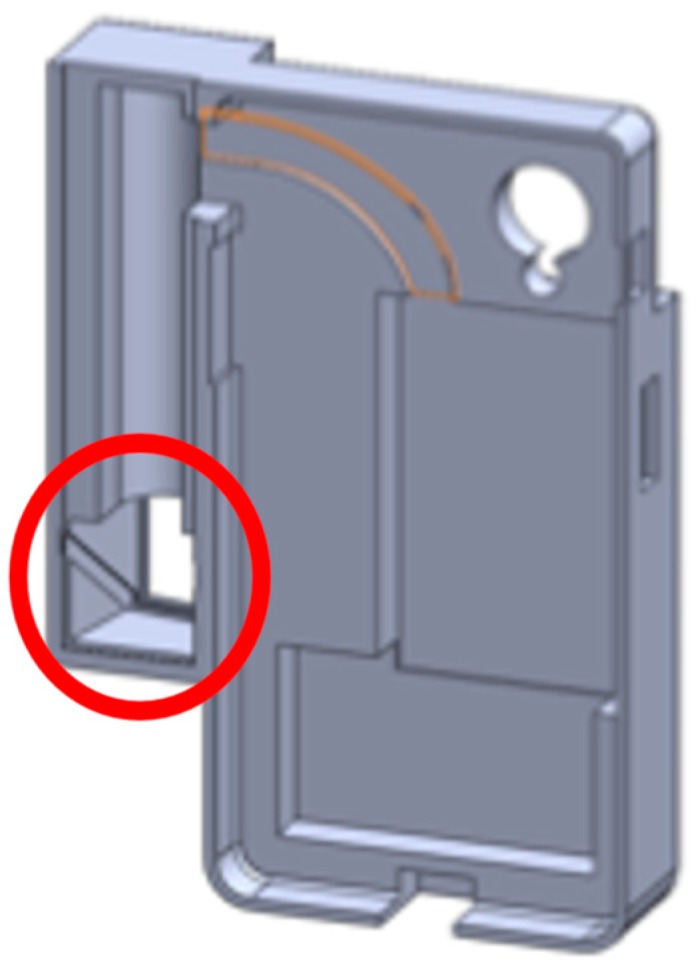
DDRS enclosure with the 2-dimensional angled mirror housing highlighted with a red circle.

**Figure 7 sensors-19-00564-f007:**
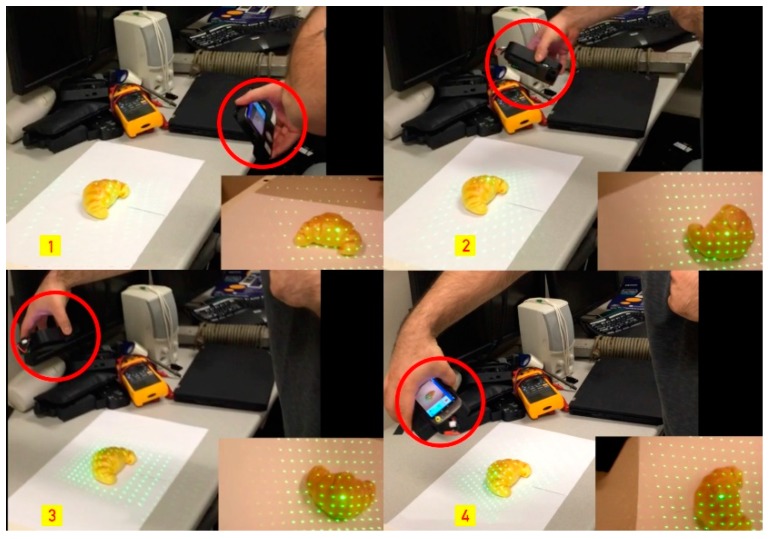
360-degree scanning of a croissant using the DDRS, which allows 360-degree scanning to get the visual information of the food from the side angles. The figure shows snapshots of the scanning process using the DDRS (circled in red) in counter-clockwise.

**Figure 8 sensors-19-00564-f008:**
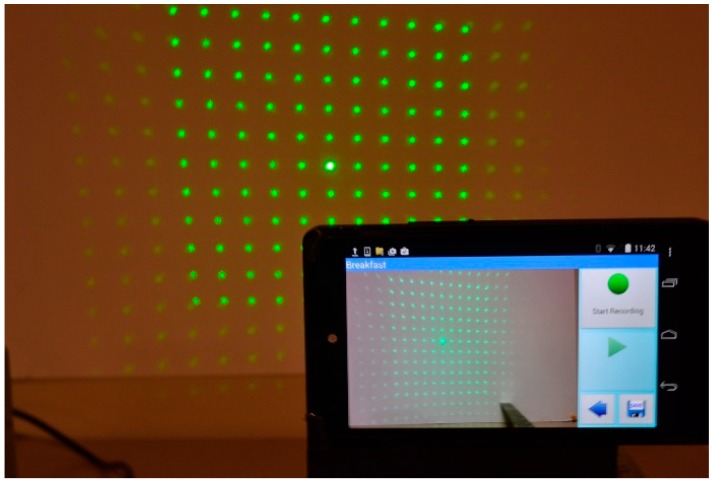
The calibration process of the DDRS consists of taking multiple snapshots of the projected laser grid onto a white flat surface at 1 cm increments from 60 cm to 10 cm. This process is only done once and allows the algorithm to have comprehensive data on pixel distance to true depth ratio, which will be used to calculate the volume.

**Figure 9 sensors-19-00564-f009:**
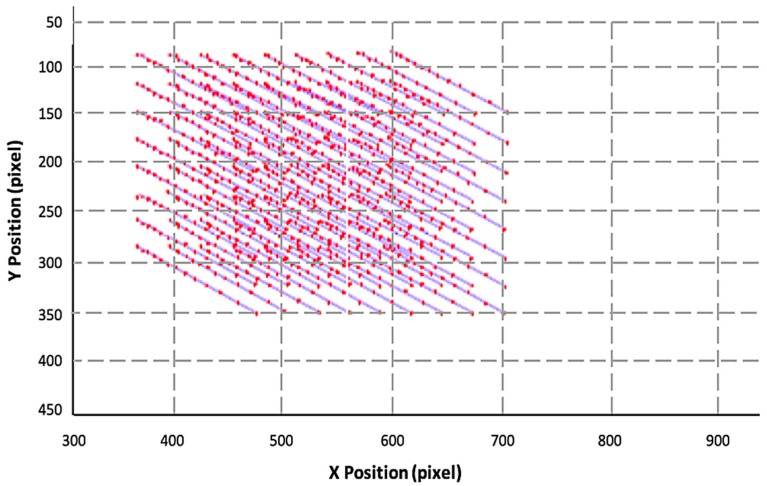
The 3D point cloud of the calibration process of a single DDRS system. The process maps out the pixel position and the actual position with respect to the DDRS camera.

**Figure 10 sensors-19-00564-f010:**
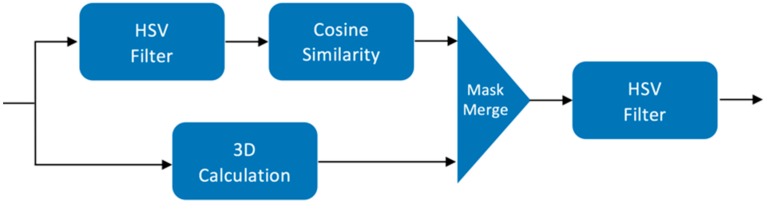
Once the frame is extracted from the scan, it is fed through our image processing algorithm to isolate the specific food item with the laser dots visible on that item.

**Figure 11 sensors-19-00564-f011:**
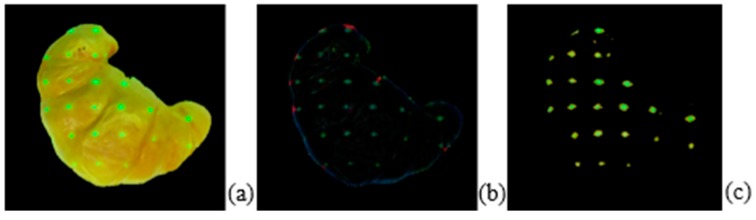
Original croissant picture (**a**), croissant picture with just the border (**b**), and croissant picture after the luminance mask is completely applied (**c**). The green dots in the image (**c**) are brighter than in image (**b**) due to the luminance mask; the luminance mask makes the dots easier to detect in the post-processing step.

**Figure 12 sensors-19-00564-f012:**
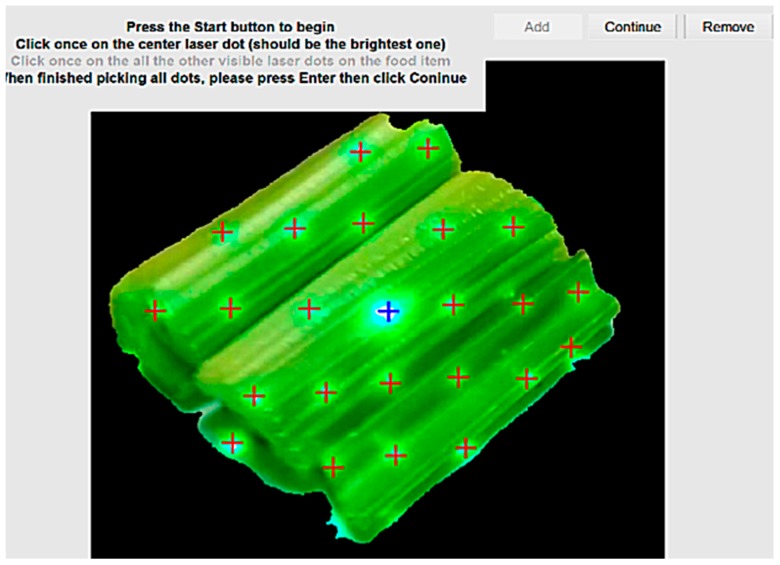
The last step, the manual dot selection GUI (which, in this case, shows a photograph of celery), corrects the automation algorithm’s errors. The picture shows a bunch of celery which has been manually traced for each laser dot to address the limitation of the algorithm in detecting green laser dots on green objects.

**Figure 13 sensors-19-00564-f013:**
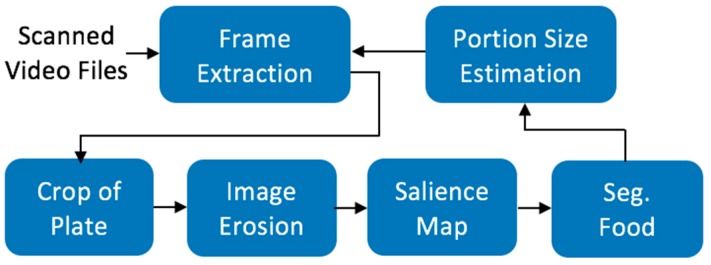
The architecture system design of the volume measurement algorithm.

**Figure 14 sensors-19-00564-f014:**
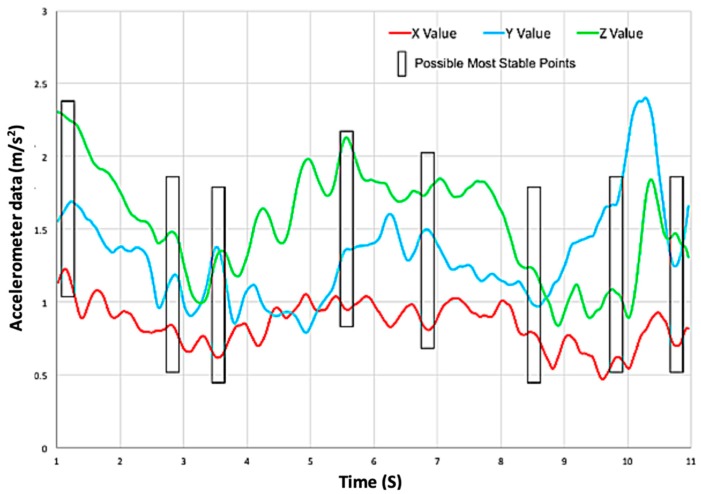
X, Y, and Z accelerometer data were taken by a first-time participant (minimum training). (The data has been shifted vertically for better demonstration).

**Figure 15 sensors-19-00564-f015:**
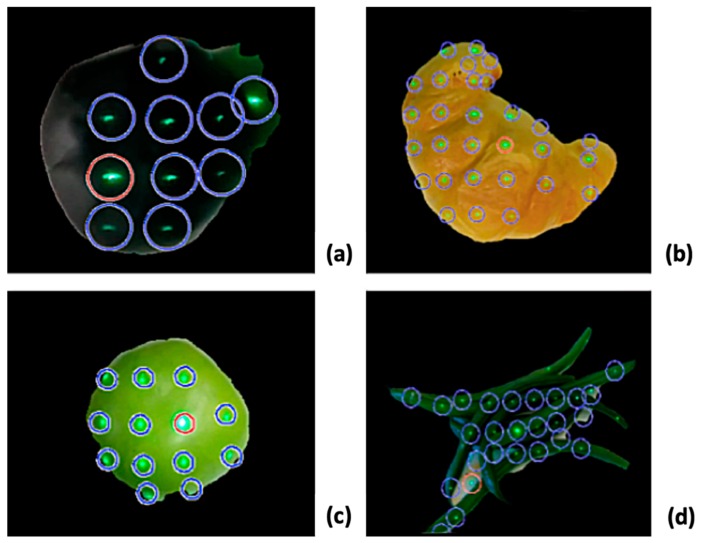
Preliminary results of the automatic dot selection algorithm and edge detection show the ability to crop and detecting dots on round, sharp, and irregular shapes of food, such as: (**a**) a pear, (**b**) a croissant, (**c**) an apple, and (**d**) green beans.

**Figure 16 sensors-19-00564-f016:**
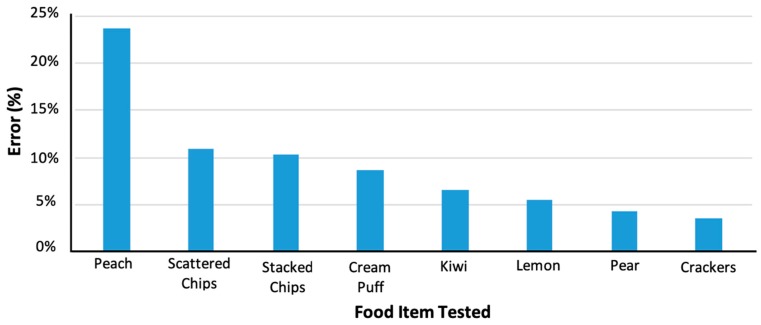
Percentage error for the number of dots detected when compared to manual dot detection.

**Figure 17 sensors-19-00564-f017:**
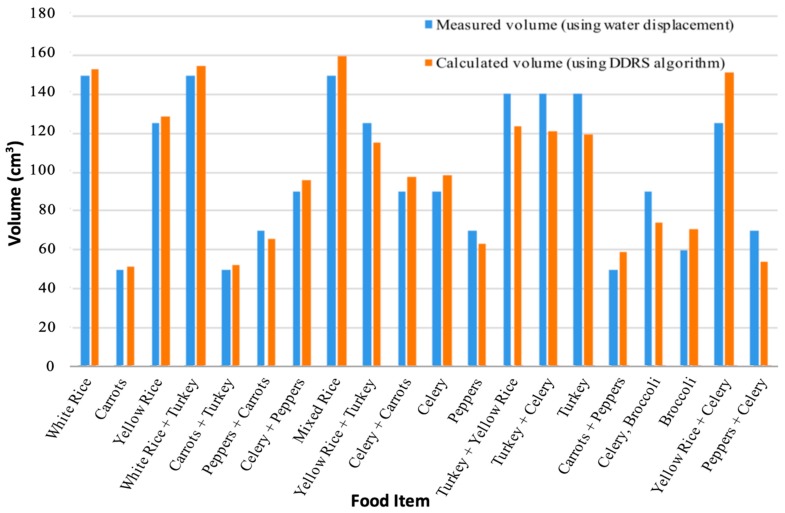
Measured volume using the DDRS algorithm vs. the calculated volume using water displacement.

**Figure 18 sensors-19-00564-f018:**
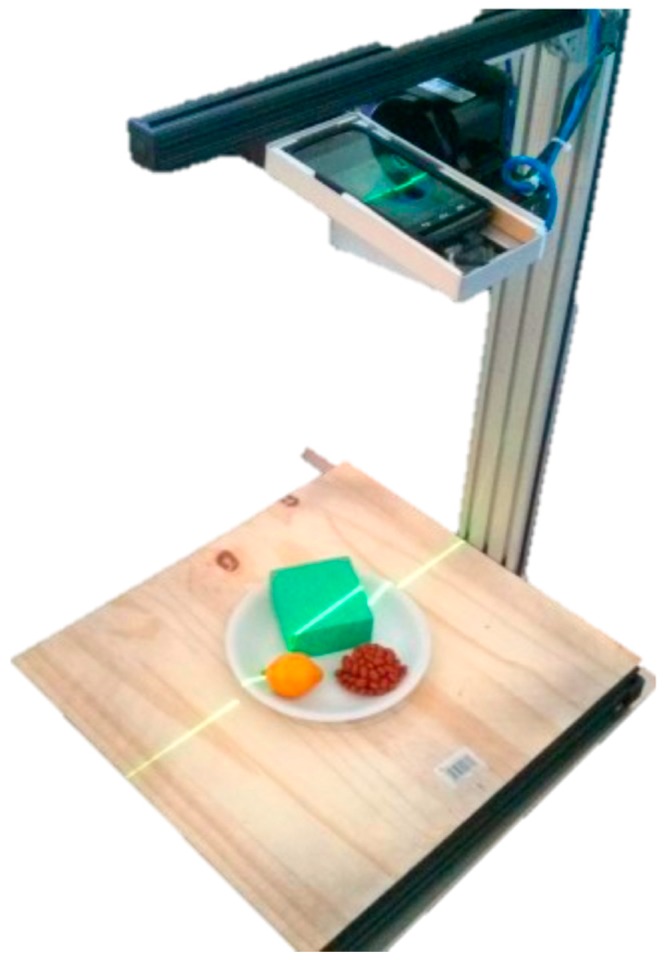
Line laser 3D scanner developed by Illionix LLC. The system uses a fixing apparatus and a smartphone which attaches to laser module hardware.

**Figure 19 sensors-19-00564-f019:**
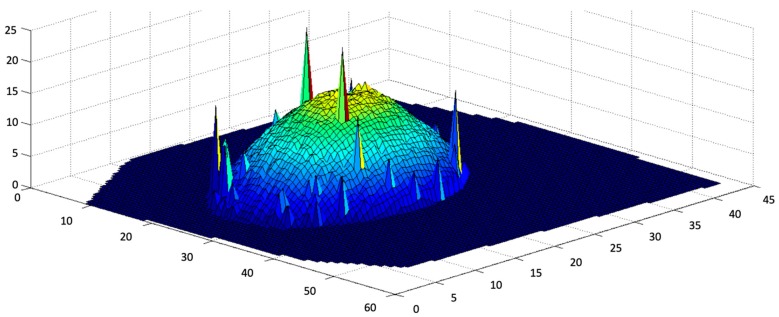
The 3D reconstructed image of the mango using the line laser scanner developed by Illionix LLC. The error spikes on the top surface of the mango can be eliminated by error catching; however, this system is limited in scanning the sides of the object since the scanner only scans from the top view angle.

**Figure 20 sensors-19-00564-f020:**
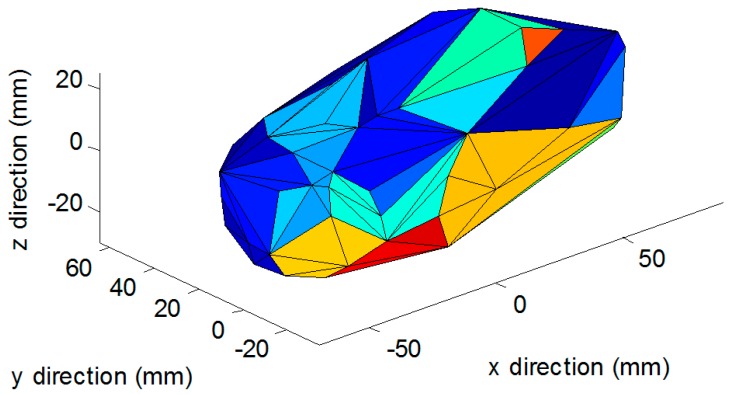
The 3D reconstructed image of the same mango, this time using the DDRS scanner. The 10% improvement is better seen in the 3D model when compared to the line laser scanner. The sides of the mango are detected and successfully reconstructed. The overall error of this mango compared to its actual volume is under 8%.

**Table 1 sensors-19-00564-t001:** Comparison between the current food volume estimation methods and the DDRS.

	Pittsburgh and Cheng Kung University [7]	DietCam [8]	Purdue University [9]	University of Calgary [10]	DDRS
**Depth Measurement Methodology**	Estimated using previous calibration—no external hardware	Using a known object―credit card	Using a known object―an index card	Microsoft Kinect as an SLS hardware add-on	Customized SLS hardware add-on
**Controlled Testing Environment**	Yes	Yes	No	Yes	No
**Percent Accuracy to the Actual Volume**	~ 90% (limited Items)	~ 90% (limited Items)	~ 90%	93.3% (limited Items)	89%
**User Experience**	Moderate	Moderate	Moderate	Hard	Easy

**Table 2 sensors-19-00564-t002:** FLEXPOINT 532 nm green laser module specifications.

Specifications	Measurement
Wavelength	532 nm
Output power max	10 mW
Output power stability	<5% after warming up at 25 °C
Beam divergence	<1 mrad
Beam angle error	<1°
Lifetime	5000 h. (at <4 mW)
Input voltage	5–30 VDC
Operating current	<300 mA
Operating temperature	15–35 °C
Mechanical	length 57 mm, diameter = 11.5 mm
Housing	aluminum

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
