# Peer review of "A Novel Mobile Structured Light System in Food 3D Reconstruction and Volume Estimation"

_sensors, 2019, doi:10.3390/s19030564_

Round 1
Reviewer 1 Report
p.p1 {margin: 0.0px 0.0px 0.0px 0.0px; text-indent: 0.8px; font: 14.0px Georgia} p.p2 {margin: 0.0px 0.0px 0.0px 0.0px; text-indent: 0.8px; font: 14.0px Georgia; min-height: 16.0px}
Equation 1 is certainly incorrect. Should the numerator have |OL| rather than |PO|?
The preference for a green laser over IR based on the user’s ability to control pointing is clear. But the detectors are more, not less sensitive to green and so stating in line 205 that it provide better visibility for the detector is incorrect.
line 329. The subject of “high volume low density” subjects is important but is dismissed too cursorily. Potato chips are certainly an example. In fact, the celery in Fig 10 is also one as the gaps due to the curved shape of the stalks creates voids. None of the top surface measurement approaches can deal with this or with gaps between the bottom surface and the reference plate - a fatal flaw. Using a statistical adjustment introduces large potential errors. To state that the authors will address this in a future “nearly completed” paper is naive.
The dot detection, triangulation and calculation of elevation is straightforward and not controversial. But the discussion of error sources is overly brief and rather dismissive. Also, comparison to more robust approaches such as measuring portion weight should be included.
Author Response
The authors would like to thank reviewer 1 for their feedback. The response has been attached as a pdf file

Reviewer 2 Report
Manuscript number: sensors-421393
Title: Novel Mobile Structured Light System in Food 3D 2 Reconstruction and Volume Estimation
Authors: Sepehr Makhsous, Hashem M. Mohammad, Jeannette M. Schenk 3, Alexander V. Mamishev and Alan R. Kristal
The application of mobile structured light system in food 3D reconstruction is an interesting task. I agree with the concepts explained in the paper. However, the paper lacks some matters to establish the viability of the proposed method. In the paper, the fundamentals of the food dietary are well described. But, other sections should be improved. Therefore, comments should be included in the manuscript.
1.- The main aim of the paper is the three-dimensional reconstruction of food. However, the paper does not provide results of the three-dimensional reconstruction of food. Result of 3D food should be included.
2.- The proposed system is performed by projecting a point matrix. This method does not provide a whole surface volume. What is the approach respect to the methods based on laser line projection, which provide the whole food surface [1]. Comments about these matters should be included.
3.- The calibration is mentioned in the paper, it is very important to determine the 3D surface [2]. However, the paper does not include the procedure to determine the vision parameters. Comments about these matters should be included.
4.- The percent accuracy is mentioned, but the method to determine this parameter is not described. Comments about these matters should be included.
[1] J. of Modern optics, Vol.63 No.13 , p.1219-1232, (2016).
[2] Sensors, Vol. 10, No.8 p. 7681-7704 (2010).

Author Response
The authors would like to thank reviewer 2 for their feedback. The response has been attached as a pdf file

Reviewer 3 Report
This work uses a mobile Structured Light System to measure the food volume and portion size of a patient’s diet in daily living conditions. This is an interesting work. The following suggestions can be considered for improving this manuscript.
1. In the result section, please provide some experimental backgrounds.
2. I have a question. Whether does the laser point profile influence the measurement accuracy?
3. What position determination method is used in your system?
4. What is your calibration accuracy? The calibration accuracy is how to influence the measurement accuracy? Please discuss them.
5. You point out that the improvement of measurement accuracy can achieve 40%. However, the results don’t show this result. Please show how to achieve 40%.
Author Response
The authors would like to thank reviewer 3 for their feedback. The response has been attached as a pdf file

Reviewer 4 Report
I suggest you closer attention to paper design and editing. The numbers of some graphic ones are not correctly brought in the text (line 347, 351, 353 and line 364, 390). It is preferred to arrange the images and the graphs to beginning or end of relevant section, to ease a continue reading. On substance, the concept of image segmentation and volume measurement, which may need further attention. I suggest to explain them in separate paragraphs (line 224), because volume measurement algorithm is not satisfactorily explained. The detection of bottom surface need further explaining (line 358 - 363). I suggest to compare between photogrammetry method and DDRS. Is it possible to use photogrammetry to improve the DDRS system?
Finally, these are my other suggestions for the necessary revision: to detail the volume estimation algorithm; verify the accuracy; in addition to a comparison with the method called manual, also provide a test with photogrammetry; etc.
Author Response

(The authors gave the same response as above.)

Round 2
Reviewer 2 Report
The paper has been corrected in good form.
Reviewer 3 Report
The authors solved my questions.
Reviewer 4 Report
Thanks for the version revised of the manuscript. It complies with the comments. I think there are a number of good starts to investigate, overall the paper is still interesting and well done: now the article dispel uncertainty on the procedure and outcome results of the DDRS.